# P2Y_12_ Purinergic Receptor and Brain Tumors: Implications on Glioma Microenvironment

**DOI:** 10.3390/molecules26206146

**Published:** 2021-10-12

**Authors:** Fernanda Bueno Morrone, Pedro Vargas, Liliana Rockenbach, Thamiris Becker Scheffel

**Affiliations:** 1Laboratório de Farmacologia Aplicada, Escola de Ciências da Saúde e da Vida, PUCRS, Porto Alegre 90610-001, RS, Brazil; pedro.vargas@acad.pucrs.br (P.V.); lilarockk@gmail.com (L.R.); mih.scheffel@yahoo.com.br (T.B.S.); 2Programa de Pós-Graduação em Biologia Celular e Molecular, PUCRS, Porto Alegre 90610-001, RS, Brazil; 3Programa de Pós-Graduação em Medicina e Ciências da Saúde, PUCRS, Porto Alegre 90610-001, RS, Brazil

**Keywords:** purinergic signaling, P2Y_12_, glioma, tumor microenvironment, platelets

## Abstract

Gliomas are the most common malignant brain tumors in adults, characterized by a high proliferation and invasion. The tumor microenvironment is rich in growth-promoting signals and immunomodulatory pathways, which increase the tumor’s aggressiveness. In response to hypoxia and glioma therapy, the amounts of adenosine triphosphate (ATP) and adenosine diphosphate (ADP) strongly increase in the extracellular space, and the purinergic signaling is triggered by nucleotides’ interaction in P2 receptors. Several cell types are present in the tumor microenvironment and can facilitate tumor growth. In fact, tumor cells can activate platelets by the ADP-P2Y_12_ engagement, which plays an essential role in the cancer context, protecting tumors from the immune attack and providing molecules that contribute to the growth and maintenance of a rich environment to sustain the protumor cycle. Besides platelets, the P2Y_12_ receptor is expressed by some tumors, such as renal carcinoma, colon carcinoma, and gliomas, being related to tumor progression. In this context, this review aims to depict the glioma microenvironment, focusing on the relationship between platelets and tumor malignancy.

## 1. Introduction

Gliomas are malignant brain tumors that are frequent in adults, characterized by a high degree of proliferation and invasion [1]. Increasingly, the glioma malignancy has been attached to the multiple mechanisms present in the tumor microenvironment (TME) associated with an evasion of immune attack. Glioblastoma multiforme (GBM) is the most aggressive grade of gliomas and is considered incurable, the median survival being only 15 months after diagnosis and recurrence being frequently mentioned [2,3]. Although chemotherapy has a therapeutic scope and considerable penetration in the blood-brain barrier, temozolomide resistance has been broadly described in GBM patients [4,5]. 

The analysis of the TME, where neoplastic and non-neoplastic cells interact, has been shown to be fundamental in understanding cancer. In addition to the role in tumorigenesis, the TME is recognized as a regulator of tumor progression and therapeutic efficacy in patients with primary and metastatic brain tumors [6]. It is known that hypoxia as well as chemo- and radiotherapy generate cell damage and promote the release of nucleotides, such as adenosine triphosphate (ATP) and uridine triphosphate (UTP), into the extracellular space [7,8]. Purinergic signaling has been proven to have a significant effect on tumor evolution since it impacts not only the tumor itself but also the TME [9]. The purinergic system in the TME is complex and plays a role in balance with immune signaling pathways. The half-life of extracellular ATP (eATP) is short because of the quick action of ectoenzymes that convert ATP to adenosine diphosphate (ADP) and then into adenosine, leading to a rich medium for growth-promoting and immunomodulatory factors [10,11,12,13]. 

In turn, ATP and other nucleotides have already been related to increased drug resistance by exerting inflammatory and pro-invasive functions acting on P2 purinergic receptors [14,15,16]. Several types of P2 receptors were shown to be involved in regulating the proliferation of numerous tumor cells, including P2X3, P2X4, P2X5, P2X7, P2Y_1_, P2Y_2_, P2Y_4_ P2Y_11_, and P2Y_12_ [17,18,19,20,21,22,23,24]. Specifically, the P2Y_12_ is a G-protein-coupled receptor that is activated by ADP and is expressed in gliomas by platelets and microglia, besides the tumor [25,26,27]. Solid tumors secrete more ATP and ADP in relation to normal tissues, and the activation of platelets by ADP-P2Y_12_R engagement directs the activation of growth factors, such as the oncogenic epidermal growth factor receptor (EGFR), to support tumor progression [28,29,30].

Since the findings demonstrate that P2Y_12_R is an important contribution to tumor growth, this review aims to describe the complexity of the glioma microenvironment, focusing on the P2Y_12_ receptor as a regulator of platelets and immune cells, and its implication in GBM growth. 

## 2. Glioma Microenvironment

The glioma microenvironment has been pointed out as a modulator of tumor progression and a predictor of successful therapy. Neoplastic and non-neoplastic cells, such as fibroblasts, microglia, cancer stem cells, endothelial, platelets and immune cells, coexist in the glioma adjacent space and are associated with invasiveness and angiogenesis processes [6]. In fact, cell communication in the brain is an important hallmark of gliomas [31]. There is a complex network of cytokines and chemokines triggered by the invasive growth of GBM that directly leads to microvascular changes supporting the recruitment of immune cells and favoring tumor-associated stromal cells [31,32,33,34]. 

The tumor-associated macrophages (TAMs) are the main immune cell constituent in the GBM microenvironment, and the glioma-microglia interaction has been pointed out as a critical barrier to the tumor’s resolution [35,36]. TAMs favor a protumor microenvironment, being related to facilitating the infiltration of GBM cells in the brain and, additionally, to inducing tumor neovascularization by producing pro-angiogenic factors such as CXC-chemokine ligand 2 (CXCL2) and vascular endothelial growth factor (VEGF) [31,37,38].

The lymphocytes, other important immune cells in the TME, can give rise to either CD4^+^ or CD8^+^ cells, which could consequently originate different T helper subsets: Th1, indicating a proinflammatory response including the secretion of interleukine-2 (IL-2), interferon-gamma (IFN-γ) and tumor necrosis factor (TNF); and Th2 as a resolutive immune subset that is capable of increasing anti-inflammatory cytokines such as IL-6, among others. Notably, gliomas are recognized as ‘cold’ tumors because of the low numbers of tumor-infiltrating immune effector cells, including CD4^+^ and CD8^+^ cells, which leads to an intense and complex immunosuppressive microenvironment based on Th2 immune responses [39,40]. Furthermore, it has been described that an important type of innate specific lymphocyte, as natural killers (NK), is reduced in GBM patients [41,42]. It seems that changes in triggering and inhibiting NK receptors in multiple cancers lead to immune surveillance escape and tumor progression [43]. 

Strikingly, platelets also perform a key role in the TME [44]. Platelets are proficient in modulating the TME by releasing inflammatory mediators and growth factors, which impact the establishment of metastasis, angiogenesis, and thrombocytosis. Growing evidences reveal platelets as key targets for cancer treatment [45,46]. It is known that patients suffering from GBM exhibit both circulating tumor cells and platelets carrying glioma-derived RNA. According to Kuznetsov et al., tumor cells could improve platelet reactivity and ‘educate’ platelets into stimulating thrombopoiesis and tumor progression [47,48].

Glioma microenvironments have a subset of cells with stem-like characteristics called glioblastoma stem cells (GSCs), which are able to drive tumor growth and to provide resistance to conventional therapy [49,50,51]. These stem-like cells are a proportion of cells within the TME that exhibit a self-renewal ability and differentiate into downstream lineages. Interactions among GSCs and the TME not only provide the maintenance of the stem-like status but also acquire aggressive behaviors including migration and invasion phenotypes [50,52,53]. 

It is known that the GBM microenvironment is rich in signaling pathways [12]. Extracellular vesicles (EVs) play a role as mediators of intercellular communication in the TME, which are able to modulate immune responses; however, their function in tumor growth is not clear [54]. In fact, it was demonstrated that glioma cells release microvesicles which can stimulate angiogenesis and promote tumor growth in a human glioma cell line [55]. Otherwise, a recent study has shown that glioma-derived extracellular vesicles (GEVs) have antiproliferative properties and reduced regulatory T cells in the glioma microenvironment [54]. 

Glioma cells are able to secrete numerous mediators such as chemokines, cytokines and growth factors that induce the permeability of astrocytes, endothelial cells, and immune cells including microglia, TAMs, myeloid-derived suppressor cells (MDSCs), effector T lymphocytes and regulatory T lymphocytes (Tregs) [56,57]. The tumor growth factor-*β* (TGF-*β*) is a cytokine found at high levels in the serum of GBM patients and has been implicated as the main factor inducing Tregs [58]. The enhancement of immune escape mechanisms is essential for tumorigenesis and can require the induction of immunosuppressive cells. Four types of changes, including tolerance, asthenia, exhaustion and senescence, were reported in the TME [59]. In physiological conditions, the role of Tregs is to migrate into inflammatory sites and inhibit an exacerbated immune response by suppressing various effector cells, including T CD4^+^ and CD8^+^ cells. The presence of Tregs also enhances the immunosuppression in the glioma microenvironment through the continuous activation of inhibitory immune checkpoints, such as antigen 4 associated with cytotoxic T lymphocytes (CTLA-4) and programmed cell death receptor-1 (PD-1) [60,61]. Unlike this, the depletion of Tregs showed a successful reversal role for effector T cells, leading to a much more favorable immune environment to attack the tumor [62]. 

Hypoxic areas and the inflammation present in the TME can represent a constant source of ATP [63,64,65]. Furthermore, the glioma treatment has been shown to be involved in increased levels of purine and pyrimidine metabolites, particularly in resistant glioma cells [66]. The extracellular ATP promotes immune responses by acting on P2 purinergic receptors expressed on both tumor and host cells and also supports an immunosuppressive and proangiogenic environment around the tumor by the generation of adenosine [12,67,68].

## 3. P2Y_12_ and Cancer

Extracellular purines (ATP, ADP and adenosine) act as endogenous signaling molecules, exerting effects on the inflammatory and immune response, neurotransmission, muscle contraction, platelet aggregation, pain, and modulation of cardiac function, among others [69]. The balance of nucleotides and nucleosides in the TME is maintained by the action of ectonucleotidases. The NTPDase1/CD39 (encoded by *ENTPD1* gene) and ecto-5′-nucleotidase/CD73 (encoded by *NT5E* gene) are the main enzymes representing the source of adenosine in the extracellular space. In the glioma microenvironment, while CD39 hydrolyzes ATP to ADP and adenosine monophosphate (AMP), being primarily expressed by immune cells, CD73 is present in glia (astrocytes, oligodendrocytes and microglia) and tumor cells and lastly converts AMP to adenosine [70,71,72,73].

Nucleotides and nucleosides in the extracellular space activate two main families of purinergic receptors: P1, a type of G protein-coupled receptor selective for adenosine, and P2 receptors that have a high affinity for di- and triphosphated nucleosides (ATP, ADP, UTP, and UDP) [9]. P2 receptors are still divided into P2X (ionotropic receptors) and P2Y (metabotropic receptors). P2X are ligand-gated ion channels expressed in all the living cells and tissues of vertebrates. They are classified in P2X1 to P2X7 [63], and each subunit has distinct pharmacological and/or physiological properties [74]. 

So far, eight subtypes of P2Y receptors (P2Y_1,2,4,6,11,12,13,14_) have been described. These G protein-coupled receptors have seven transmembrane domains, with the N-terminal tail facing the extracellular environment and the C-terminal tail facing the intracellular environment [75]. The P2Y receptors have been divided into two structurally distinct subgroups: the G_q_ protein-coupled P2Y, which includes the P2Y_1_, P2Y_2_, P2Y_4_, P2Y_6_, and P2Y_11_ receptors, and the G_s_/G_i_-coupled P2Y, including the P2Y_12_ P2Y_13_, and P2Y_14_ receptors. The G_q_ protein-coupled P2Y receptors activate phospholipase C and calcium signaling, while the G_s_/G_i_-coupled P2Y receptors affect adenylyl cyclase (AC), which catalyzes the synthesis of cyclic AMP (cAMP), causing an alteration in the intrinsic cell metabolism [63,75,76,77].

The P2Y_12_R is a chemoreceptor for ADP mainly found on the platelet surface [78,79]. ADP can be derived from an exogenous source or released by activated platelets themselves and acts at first by stimulating the P2Y_1_ receptor, which is responsible for beginning the platelet aggregation, and afterwards in P2Y_12_, amplifying the aggregation response [80]. Platelets interact with thromboxane A2, collagen, and thrombin in addition to ADP, causing intracellular signaling that activates the fibrinogen receptor. The coupling of the fibrinogen-fibrinogen receptor in platelets results in platelet accumulation that detains bleeding at the wound site [78]. The activation of P2Y_12_R contributes to the stabilization of thrombotic events in a manner that is dependent on the phosphorylation of phosphoinositide-3-kinase (PI3K)/Akt pathways [29,81]. 

The P2Y_12_R expression is found to be relevant in tumor cells. Some cancers have already shown P2Y_12_R expression; nonetheless, the receptor role in tumor development is not completely understood [82,83,84,85]. According to Elaskalani et al., pancreatic cancer cells could express the functional P2Y_12_R and exhibited a potential ADP-dependent cell proliferation by promoting EGFR and Akt signaling in vitro [30]. P2Y_12_R was also shown to enhance withstanding chemotherapy-induced cytotoxic stress in breast cancer [82], while the P2Y_12_R antagonism was shown to reduce pancreatic tumor growth in synergism with chemotherapeutic agents [30]. Recently, Sharma et al. demonstrated the P2Y_12_R-dependent mechanisms of cell migration and invasion in cervical cancer [85]. Similarly, in glioma cells, ADP-P2Y_12_ stimulation induced extracellular-signal-regulated kinase (ERK1/2) and PI3K pathways leading to tumor proliferation [83]. 

Additionally, the functional P2Y_12_R can perform a role in immune cells within the TME. An interesting study showed a positive correlation among P2Y_12_ expression and the infiltration of a non-functional and immunosuppressant cell phenotype; at the same time, P2Y_12_ was negatively correlated to a low amount of effector immune profile such as activated NK cells [86]. Besides, in vivo experiments demonstrated that P2Y_12_R deficiency can modulate microglial function by inhibiting the chemotaxis of these cells to focal injury [87].

Notably, P2Y_12_R is closely related to the development and progression of different types of cancer; regarding this, the pharmacological inhibition of this purinergic receptor stands out as a new route for reducing tumor proliferation (Figure 1). Although inhibitors of platelet aggregation, such as clopidogrel, were originally developed to treat thrombotic accidents [88], they show a great potential to be a repositioned drug for cancer therapy. Some current clinical trials including P2Y_12_R antagonists in cancer are listed below (Table 1). Interestingly, the outcomes from the NCT00263211 terminated study have shown reduced circulating cancer cells in metastatic breast cancer patients who have received clopidogrel and aspirin.

## 4. Platelets, P2Y_12_, and Gliomas

Platelets, beyond the action in hemostasis and thrombosis, have been related to tumor growth and metastasis. In fact, tumor cells can activate platelets [89,90]. It is known that tumor cells have a procoagulant activity, and several studies have demonstrated that neoplastic transformation can stimulate coagulating pathways, both in vitro and in vivo [91,92,93]. The main cellular initiator of coagulation, tissue factor (TF), is overexpressed in a range of cancers including gliomas, and likely supports a thrombotic environment. Two alterations have been cited as being responsible for TF upregulation: phosphatase and tensin homolog (PTEN) loss and hypoxia [91]. The upregulation of TF in GBM was also correlated to mutations in the EGFR and the loss of E-cadherin, which is considered an essential event in the epithelial-mesenchymal transition (EMT) [91,94]. Besides, the platelet-tumor interactions cause an upregulated expression of other mesenchymal markers, such as matrix metalloproteinase-9 and vimentin, expanding the invasive signature in tumors [95]. The invasion of the surrounding parenchyma is one of the hallmarks of the GBM that impacts directly on patient survival [96]. 

In agreement with platelets’ involvement in cancer malignancy, a study with a retrospective cohort of GBM patients treated with surgery plus chemoradiation showed that those patients with high levels of blood platelets after treatment had a decreased survival compared with those with low-platelet blood levels, 11 and 28 weeks, respectively [97]. Furthermore, aspirin-induced apoptosis in glioma cell lines and in tumor-bearing nude mice also led to a reduction in animals’ tumor volume [98]. Besides that, aspirin enhanced temozolomide effects [99], and patients treated with aspirin and clopidogrel showed a declined risk of cancer development [100].

Activated platelets (ADP-P2Y_12_) secrete α-granules that contain a variety of mediators and chemokines, these being important regulators of leukocyte migration into the TME [101]. Studies have demonstrated that the CXCL12 released by platelets induces leucocyte recruitment to the tumor site, such as macrophages, stimulating tumor growth and angiogenesis. Macrophages in the glioma microenvironment are “plastic” and frequently shape into a tumor-associated phenotype, being a barrier to immune attack [12]. The expression of endothelial-leucocyte adhesion molecules is also induced by activated platelets expressing interleukin 1 beta (IL-1β) [102]. Significantly, the immunosuppressive TGF-β is secreted by activated platelets and supports the tumor escape from immune system recognition. TGF-β is involved in blocking CD4^+^ and CD8^+^ T cell activation and predominantly generates inducible Tregs. Generally, the conversion of neutrophils on a pro-tumorigenic phenotype is totally favored by the presence of TGF-β into the TME [102,103,104]. 

Cancer cells that more effectively activate platelets have been shown to have high aggressiveness rates. This is because platelets promote migration and invasion by shielding tumor cells from NK cells’ attack, enhancing adhesion and transmigration across the endothelium and promoting angiogenesis and proliferation by the release of ATP and ADP [89,90,95,105]. The nucleotides are mediators that are known to participate in tumor growth and inflammation [10,67]. ADP binding to P2Y_12_R plays a critical role in maintaining a very low cAMP level and sustained high activity of the PI3K/Akt cascade for glioma growth. P2Y_12_R activation inhibits the cAMP-induced differentiation of C6 glioma cells and converts the differentiation into an enhanced proliferation [106,107]. Elevated levels of AC activity and cAMP levels are prejudicial for tumors because they were related to the stabilization of the cell morphology and diminished growth rates in neoplastic cells, so the maintenance of a vicious cycle involving nucleotides in the TME is relevant for tumors [108].

Besides its role in tumor proliferation through activating platelets and consequently regulating leukocytes’ migration, P2Y_12_R is also involved in the tumor proliferation pathway resulting from microglia interactions. Pre-neoplastic cells interact with microglia in P2Y_12_R-dependent Ca^2+^-mediated ATP signaling, and this interaction is essential for its proliferative capacity [109]. Moreover, P2Y_12_R activated by ADP in microglial cells induces chemotaxis as a “find me” signal, while UDP operates as an “eat me” signal [110]. 

P2Y_12_R is involved in glioma proliferation, differentiation and survival, as confirmed and shown in C6 cells under serum deprivation presenting an increased P2Y_12_R expression [111]. In agreement with this, Shchors et al. showed that combining Imipramine (antidepressant agent) and ticagrelor (P2Y_12_R inhibitor) potentiated the induced antiproliferative effects, cell death and autophagy via cAMP enhancement [112].

In general, P2Y_12_R exhibits crucial roles in the maintenance of a pro-glioma environment, either through its expression in platelets or even in the tumor itself, as can be seen in the summary in Figure 2.

## 5. Conclusions

The treatment of malignant tumors is a challenge. Several protumor signaling pathways coexist in the TME and can be a deadlock for therapeutic success. In view of the involvement of P2Y_12_R in glioma malignancy, directly or indirectly through activated platelets and stromal cells, the P2Y_12_R antagonists may show a high potential as antiglioma drugs. The advantage is that the existence of available P2Y_12_R antagonists such as clopidogrel and ticagrelor, with an affordable cost and for use in patients presenting platelet disorders, may contribute to treatment accessibility through the use of drug repositioning as an interesting tool to cancer therapy. 

## Figures and Tables

**Figure 1 molecules-26-06146-f001:**
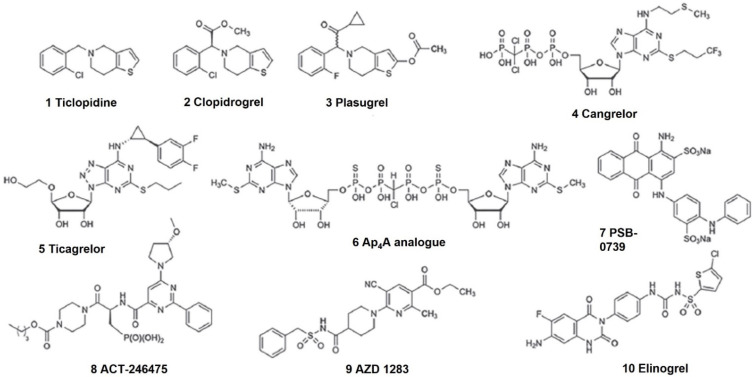
P2Y_12_ receptor antagonists (Adapted from Jacobson, 2020, with permission from John Wiley and Sons) [76].

**Figure 2 molecules-26-06146-f002:**
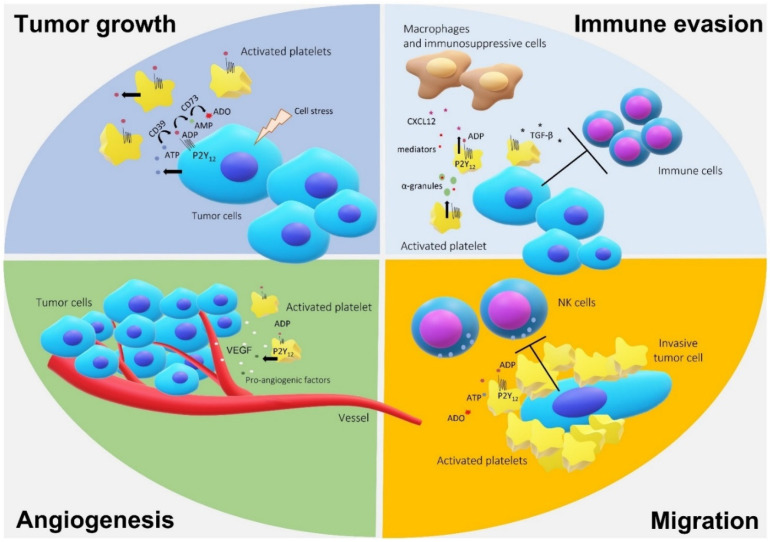
Representation of P2Y_12_ receptor and platelet participation in the glioma microenvironment. The hypoxia environment favors the secretion of adenosine triphosphate (ATP) and the conversion into other nucleotides, such as adenosine diphosphate (ADP), responsible for stimulating tumor growth, angiogenesis, and migration by P2Y_12_ activation. The release of ATP can stimulate the recruitment and activation of lymphocytes, monocytes and macrophages that favor the growth of tumor cells while secreting proinflammatory interleukins. Platelets are recruited and can protect tumor cells from the immune system.

**Table 1 molecules-26-06146-t001:** Summary of current clinical trials for P2Y_12_ antagonists in cancer.

Agent	Other Combination Target	Clinical Trial Identifier	Phase	Status
Clopidogrel	No	NCT02404363	Phase III in locally advanced or metastatic pancreatic cancer	Terminated (Recruitment problems)
Clopidogrel	Aspirin	NCT00263211	Phase II in metastatic breast cancer	Terminated (Low percentage of patients with detectable circulating cancer cells at baseline)
Clopidogrel	Acetyl salicylic acid and Alvocidib	NCT00020189	Phase II in recurrent/metastatic squamous cell carcinoma of the head and neck	Completed (No results posted)
Clopidogrel	Acetyl salicylic acid and Pembrolizumab	NCT03245489	Phase I in recurrent or metastatic squamous cell carcinoma of the head and neck	Recruiting
Clopidogrel	Aspirin	NCT00940784	Phase II in Polycythemia Vera	Withdrawn

## Data Availability

Not applicable.

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
