# Peer review of "P2Y12 Purinergic Receptor and Brain Tumors: Implications on Glioma Microenvironment"

_molecules, 2021, doi:10.3390/molecules26206146_

Round 1
Reviewer 1 Report
This interesting minireview summarizes the role of P2Y12 receptor in gliomas, specially about its role in tumor microenvironment. It is well written and easy to follow, I have some suggestions/comments:
Line 51: P2X3, P2X4 and P2Y4 receptors have to be added to that list, for references you can check this recent review: doi: 10.1007/s11302-021-09785-8
Line 140: Phrasing is confusing, because it first states that all P1 and P2 receptors are G protein coupled and P2 are composed by P2X and P2Y receptors (as correctly stated later). Please rewrite the first sentence to avoid this confusion.
Line 148: Again, phrasing is not clear please clearly separates the two categories: i) P2Y receptors coupled to Gq and PLC-Ca2+ signaling and ii) P2Y receptors coupled to Gs/Gi and regulating AC-cAMP signaling.
Line 151: Related to the previous comment, it will useful to the reader to state the pathway associated with P2Y12 receptor (I understand that is Gi-AC-cAMP)
Line 188: I found very interesting this information. Are there more details about the studies of table 1? (For example, if the ones that are already terminated are will be available soon for treatment)
Author Response
Response to Reviewer 1 Comments
Point 1: Line 51: P2X3, P2X4 and P2Y4 receptors have to be added to that list, for references you can check this recent review: doi: 10.1007/s11302-021-09785-8
Response 1: We thank you very much for the comments. As suggested, we have included the P2X3, P2X4, and P2Y4 receptors (line 51). Please, note that we cited three new references in the final version of the manuscript: doi:10.1152/ajplung.00447.2002; doi:10.18632/oncotarget.6240; doi:10.1111/j.1474-8673.2007.00389.x.
Point 2: Line 140: Phrasing is confusing, because it first states that all P1 and P2 receptors are G protein coupled and P2 are composed by P2X and P2Y receptors (as correctly stated later). Please rewrite the first sentence to avoid this confusion.
Response 2: We thank you and agree with your comment. We have reformulated the sentence about P1 and P2 receptors (lines 148-152). We hope that the phrase is more understandable now.
Point 3: Line 148: Again, phrasing is not clear please clearly separates the two categories: i) P2Y receptors coupled to Gq and PLC-Ca2+ signaling and ii) P2Y receptors coupled to Gs/Gi and regulating AC-cAMP signaling.
Response 3: We agree with your comment. We have reformulated the explanation about P2Y receptors (lines 158-164). We thank you and hope that the new sentence is clear.
Point 4: Line 151: Related to the previous comment, it will useful to the reader to state the pathway associated with P2Y12 receptor (I understand that is Gi-AC-cAMP)
Response 4: We thank you and agree with your comments. We have organized the P2Y receptors into their respective pathway-related subgroups (lines 158-164). We understand that your suggestion is included in this context.
Point 5: Line 188: I found very interesting this information. Are there more details about the studies of table 1? (For example, if the ones that are already terminated are will be available soon for treatment)
Response 5: We thank you for your observation. We have included a new paragraph about the concluding results available (lines 201-203), and we have added some comments about the status of all studies cited in Table 1.
Reviewer 2 Report
The manuscript by Morrone et al.,. presents a comprehensive depiction of the role of P2 purinergic receptors in gliomas, the most common malignant brain tumors in adults. Specifically, the role of platelets through ADP-P2Y12 engagement in cancer context is elucidated. In addition, the P2Y12 receptor expression in renal carcinoma, colon carcinoma, and gliomas is reported to be related to tumor progression. The review is of interest and well-designed. The issue of drugs repositioning as tool to cancer therapy is always a great opportunity to fight cancer. Well done.
Author Response
Reviewer 2
The manuscript by Morrone et al... presents a comprehensive depiction of the role of P2 purinergic receptors in gliomas, the most common malignant brain tumors in adults. Specifically, the role of platelets through ADP-P2Y12 engagement in cancer context is elucidated. In addition, the P2Y12 receptor expression in renal carcinoma, colon carcinoma, and gliomas is reported to be related to tumor progression. The review is of interest and well-designed. The issue of drugs repositioning as tool to cancer therapy is always a great opportunity to fight cancer. Well done.
Response: We would like to thank you for the valuable comments about our paper, and we take this opportunity to thank your relevant comments.
Reviewer 3 Report
MS No: Molecules
MS Title: P2Y12 Purinergic Receptor and Brain Tumors: Implications on 2 Glioma Microenvironment
Article Type: Review
Morrone et al. aim to depict the glioma microenvironment with a focus on the relationship between platelets and tumor malignancy. The tumor-associated microenvironment frequently riches in growth-promoting and immunomodulatory signals, thereby contributing to tumor aggressiveness. In response to hypoxia and glioma-related therapy, amounts of ATP and ADP strongly increase in the extracellular space to trigger the purinergic signaling by nucleotides binding onto the P2 receptors. Intriguingly, several cell types in the tumor microenvironment bear the P2 receptors and are supposed to facilitate tumor growth. Furthermore, tumor cells themselves can activate platelets by the ADP-P2Y12 engagement, protecting tumors from the immune attack and providing molecules that contribute to their growth and also maintenance of a rich environment to sustain the protumor cycle. This review is attractive.
Minor comments:
- Please carefully recheck the abbreviations following after the first-time use such as epidermal growth factor receptor (EGFR).
- Please provide more information about the ADP, cAMP, ATP, and AMP levels in response to various glioma treatments as well as corresponsive aggressive behaviors in clinic.
- Please provide some other information about current clinical trials targeting platelets in cancers including glioma. Otherwise, preclinical studies for P2Y12 antagonists are also welcome.
Author Response
Response to Reviewer 3 Comments
Point 1: Please carefully recheck the abbreviations following after the first-time use such as epidermal growth factor receptor (EGFR).
Response 1: We thank you for your observation. We have checked the abbreviations and correct them as follow:
We added the abbreviation AC (lines 162 and 250);
We removed the extended name of EGFR (line 179);
We added the full name Interleukin 1 beta (line 235).
Point 2: Please provide more information about the ADP, cAMP, ATP, and AMP levels in response to various glioma treatments as well as corresponsive aggressive behaviors in clinic.
Response 2: We thank you for your observation. In an attempt to correspond with your solicitation, we included a phrase (lines 130-132) “Furthermore, the glioma treatment has been shown to be involved in increased levels of purine and pyrimidine metabolites, particularly in resistant glioma cells [66]”.
Point 3: Please provide some other information about current clinical trials targeting platelets in cancers including glioma. Otherwise, preclinical studies for P2Y12 antagonists are also welcome.
Response 3: We thank for the suggestion.
Regarding glioma treatments targeting platelets, we added a paragraph in the new version of the manuscript (lines 220-227). We also have included a new paragraph about the concluding clinic results available (lines 201-203) and we add comments about the status of all clinical studies in table 1. Moreover, you can find preclinical studies data about the effect of P2Y12 antagonists in cancer (lines 177-182) and in glioma (lines 261-265). Finally, we would like to highlight that many works cited in the paragraph about platelets, P2Y12, and gliomas (lines 246-259) have used P2Y12 antagonists as a treatment to prove the involvement of the receptor in the studied mechanisms.
Reviewer 4 Report
This review by Morrone and colleagues on the P2Y12 Purinergic Receptor and Brain Tumors is interesting and clearly written.
However, some improvements should be included. My suggestion as follows:
- Clinical Trials based on drug repositioning of platelet inhibitors, stated in Table 1, should be discussed more extensively in the text. Just few essential information: -did the completed one reach the endpoints? Why two of them are terminated? - Did the trials give additional information on biological pathways in tumor/TME upon treatments? (Analysis on patient biological fluids or similar traslational investigations)
- Although the Review is not focused on Extracellular Vesicles, I would suggest to mention them in the paragraph "Glioma Microenvironment". Extracellular Vesicles are important mediators of information in cancer progression,: authors should take EVs in account when they give an overview of cell to cell communication in glioma, and take a look to the possible involvement of P2Y in the release of EVs from platelet.
- From line 73 to line 77 the sentence is not clear. Please, rephrase it.
- This review cites too many reviews. Please, try to cite the original works instead.
Author Response
Response to Reviewer 4 Comments
Point 1: Clinical Trials based on drug repositioning of platelet inhibitors, stated in Table 1, should be discussed more extensively in the text. Just few essential information: -did the completed one reach the endpoints? Why two of them are terminated? - Did the trials give additional information on biological pathways in tumor/TME upon treatments? (Analysis on patient biological fluids or similar traslational investigations).
Response 1: We thank you and agree with your comment about the relevance of our manuscript. Please, note that we have included more information about the status of all studies in table 1. We also have discussed the main result obtained from a terminated study (lines 201-203).
Point 2: Although the Review is not focused on Extracellular Vesicles, I would suggest to mention them in the paragraph "Glioma Microenvironment". Extracellular Vesicles are important mediators of information in cancer progression: authors should take EVs in account when they give an overview of cell to cell communication in glioma, and take a look to the possible involvement of P2Y in the release of EVs from platelet.
Response 2: We thank you for the suggestion. We have added information on extracellular vesicles as mediators in cancer progression in the section "Glioma Microenvironment" (lines 106-112).
Point 3: From line 73 to line 77 the sentence is not clear. Please, rephrase it.
Response 3: We have revised and rewrite the sentence. We hope that the phrase is more understandable now. Thank you for your comment.
Point 4: This review cites too many reviews. Please, try to cite the original works instead.
Response 4: We have revised the references and we have included original works in the final manuscript.